# Degradation of Chemical Components of Thermally Modified *Robinia pseudoacacia* L. Wood and Its Effect on the Change in Mechanical Properties

**DOI:** 10.3390/ijms232415652

**Published:** 2022-12-09

**Authors:** Adam Sikora, Kateřina Hájková, Tereza Jurczyková

**Affiliations:** Department of Wood Processing and Biomaterials, Faculty of Forestry and Wood Sciences, Czech University of Life Science Prague, Kamýcká 1176, 165 21 Prague, Czech Republic

**Keywords:** thermal modification, hardwood, black locust, static and dynamic bending, chemical analysis

## Abstract

Currently, emphasis is placed on using environmentally friendly materials both from a structural point of view and the application of protective means. For this reason, it is advisable to deal with the thermal modification of wood, which does not require the application of protective substances, to increase its service life. The main reason for the thermal modification of black locust is that although black locust grows abundantly in our country, it has no industrial use. It is mainly used outdoors, where thermal modification could increase its resistance. This article deals with the thermal modification of black locust wood (*Robinia pseudoacacia* L.) and the impact of this modification on the chemical components of the wood with an overlap in the change in mechanical properties compared to untreated wood. Static (*LOP*, *MOR*, and *MOE*) and dynamic (*IBS*) bending properties were evaluated as representative mechanical properties. At the same time, the impact of thermal modification on the representation of chemical components of wood (cellulose, lignin, hemicellulose) was also tested. As a result of the heat treatment, the mechanical properties gradually decreased as the temperature increased. The highest decrease in mechanical values found at 210 °C was 43.7% for *LOP* and 45.1% for *MOR*. Thermal modification caused a decrease in the content of wood polysaccharides (the decrease in hemicelluloses content was 33.2% and the drop in cellulose was about 29.9% in samples treated at 210 °C), but the relative amount of lignin in the wood subjected to increased temperature was higher due to autocondensation, and mainly because of polysaccharide loss. Based on the correlations between chemical and mechanical changes caused by thermal modification, it is possible to observe the effects of reducing the proportions of chemical components and changes in their characteristic properties (*DP*, *TCI*) on the reduction in mechanical properties. The results of this research serve to better understand the behavior of black locust wood during thermal modification, which can primarily be used to increase its application use.

## 1. Introduction

The current era emphasizes the renewability and processing of natural raw materials. One of the most used raw materials is wood, thanks to its renewability and workability. Although wood is a renewable source of raw material, for its optimal use it needs to be protected from many external factors. This is what various methods of its modification are used for. These methods serve mainly to preserve its positive properties (strength, flexibility, low weight, etc.) and to eliminate the negative ones (dimensional instability due to humidity, low biological resistance, flammability, etc.). Thermal modification uses high temperatures to improve the properties of wood. At high temperatures, wood changes its structure and properties, which is the purpose of its modification.

Thermal modification of wood is an environmentally friendly method of wood protection. The result of thermal modification are wooden products that have so-called added value. Thanks to thermal modification, wood may have better resistance to color changes, weather conditions, or biological pests, as well as better dimensional stability [1,2,3]. The concept of thermally modified wood dates back to 1915 as a result of research at the United States Forest Products Lab. Research and development continued into the 1990s, with further attempts at commercialization, but again without success. The process was first commercialized in the last few decades of the 20th century in Finland. In the twenty-first century, thermally modified wood is finding increasing application in exterior cladding, decking, and joinery applications and is a well-established commercial technology, with European production now exceeding 500,000 m^3^ [4]. However, there is still a lack of detailed knowledge about the effect of thermal modification on the change in the chemical components and mechanical properties of *Robinia pseudoacacia* L. wood. This is important, mainly because in our conditions, it is a relatively little-used species of wood that in the future, due to climate change, may have potentially higher usability than at present, mainly for fast-growing trees [5].

Fast-growing woody species contain a high proportion of juvenile wood, which have a negative effect on wood quality because the growth rate of these species affects the density, dimensional stability, size of annual rings, and resistance to biological pests. The growth rate of woody plants affects the density, dimensional stability, size of annual rings, and resistance to biological pests. Therefore, the wood of fast-growing trees has a low density, worse dimensional stability, and significantly lower resistance to biological damage, which are the properties that can be improved precisely by thermal modification [1,3,6].

Wood is a natural polymer composite material that consists of cellulose, hemicelluloses, lignin, extractives, and mineral substances [7,8]. During the process of thermal degradation, organic acids from hemicelluloses create an acidic environment that, combined with higher temperature, breaks down the lignin–polysaccharide bonds of wood. Thus, it transforms wood from a hygroscopic material into a hydrophobic material [9,10,11]. The hygroscopic properties of wood after thermal modification are related to changes in the chemical composition of wood, especially hemicelluloses. Thermal modification results in a reduction in OH content (mainly due to hemicellulose degradation); hence, the EMC (equilibrium moisture content) at any given RH would be expected to be lower compared to unmodified wood, assuming the primary sorption sites model to be the correct explanation for hygroscopic behavior [11]. This simple relationship between OH content and EMC, was called into doubt by the work of Rautkari et al. [12] who found that there was no direct correlation between OH content, as measured by HDX and the EMC of TMT (thermally modified timber). This led to the conclusion that there were additional mechanisms responsible for the reduction in EMC, other than the accessible OH content. Tuong and Li [1] investigated the change in the chemical structure of acacia hybrid wood after heat treatment, which took place in a nitrogen-protective environment at a temperature of 210–230 °C. Their results show that after the heat treatment, the number of hydroxyl groups and hygroscopic properties is reduced, which leads to an improvement in the dimensional stability of the wood.

This study was conducted to elucidate the chemical and mechanical changes in thermally treated black locust (*Robinia pseudoacacia* L.) wood. The research is focused on two areas of change that affect the quality and properties of wood. The first area aims to the effect of thermal modification temperature on the change of chemical components compared to untreated wood. The second area is devoted to mechanical properties such as modulus of elasticity, proportionality limit, modulus of rupture, and impact flexural strength of thermally modified and reference untreated black locust (*Robinia pseudoacacia* L.) wood samples.

## 2. Resultes and Discussion

### 2.1. Mechanical Analysis

Table 1 shows changes in selected static bending properties of thermally modified black locust wood (modulus of elasticity, limit of proportionality, and modulus of rupture) and values for impact bending strength.

### 2.2. Static Analysis

A closer look at the static bending results can be seen in Figure 1a,b and Figure 2. Figure 1a shows changes in the limit of proportionality and modulus of rupture due to thermal modification in the temperature range from 160 °C to 210 °C. The trend of changes is comparable for both properties. In the case of the limit of proportionality according to Duncan’s test, there was no statistically significant decrease due to thermal modification at 160 °C (95.7 MPa) compared to the reference (106.6 MPa), where the level of significance is *p* = 0.06. In all other cases, the significance level was *p* ≤ 0.05. The lowest limit of proportionality values was achieved during thermal modification at 210 °C (60.0 MPa), representing a decrease of 43.7% compared to the reference values. In the case of modulus of rupture, according to Duncan’s test, all changes as a result of thermal modification were statistically significant (*p* ≤ 0.05). Modulus of rupture changes due to thermal modification had a linear character due to individual degrees of thermal modification. The reference test specimens reached average values of 150.8 MPa; these values at 160 °C thermal modification (124.6 MPa) decreased by 17.4%. In the case of 180 °C thermal modification (101.4 MPa), this decrease was by 32.8% compared to the reference. The largest decrease compared to the reference was recorded for test specimens subjected to thermal modification at 210 °C (82.8 MPa), when it decreased by 45.1%. Changes in the bending properties of thermally modified wood are associated with structural changes and reduced sensitivity to moisture. Changes in specific bending properties are linked to the chemical and structural composition of the cell wall down to the molecular level. The wood cell wall is considered a natural composite material formed by the complex interactions of its main polymers, cellulose, hemicelluloses, and lignin [13,14]. Thermal modification initiates a process of polymer degradation and possible crosslinking with degradation products, which substantially changes the initial structure and interaction of wood polymers, together with anatomical changes on a macroscopic to microscopic scale, which can lead to significant changes in material properties. Increased temperature causes chemical changes in the main components of wood and extractive substances. Their range depends on the duration of action and the temperature at which the thermal modification is carried out. At temperatures between 20 °C and 150 °C, wood is dried, whereby the wood loses water, starting with free water and ending with bound water. In the temperature range of 180 to 250 °C, which is usually used for the thermal modification of wood, significant chemical changes take place in the wood.

Thermal degradation of the hemicelluloses commences at lower temperatures compared to lignin. There is also a smaller reduction in amorphous cellulose content, but a relative proportional increase in the lignin content. Because thermal modification results in the degradation of cell wall polymers, there should be no swelling associated with the production of thermally modified wood. The hemicelluloses display chain mobility when moisture is present, and their removal means that this property is reduced in thermally treated wood. With the reduction of this amorphous polysaccharidic part, the resistance to decay is improved, too. The mobility of the amorphous part of the cell wall polymer network is also restricted if crosslinking takes place. The potential formation of crosslinks under heat-treatment conditions explains the improved dimensional stability of the thermally modified wood. The wood polymer matrix then acts as a stiff, elastic, but brittle material. This change in mobility affects the mechanosorptive properties of the material. In addition, the increased crystallinity of cellulose chains in wood also contributes to the increase in strength properties [2,3].

A different trend was observed for changes in the modulus of elasticity due to thermal modification, which can be seen in Figure 1b. According to Duncan’s test, the individual levels of significance of changes in the modulus of elasticity were determined due to the individual degrees of thermal modification. The only statistically insignificant level of significance (*p* = 0.78) was demonstrated by the change in the modulus of elasticity between thermal modification at 160 °C (12,324.6 MPa) and 180 °C (12,216.3 MPa). For all remaining degrees of thermal modification, there were statistically significant changes (*p* ≤ 0.05). The highest values of the modulus of elasticity were recorded for the reference test specimens (13,269.4 MPa), while the lowest was recorded for the thermal modification at 210 °C (10,755.3 MPa), which meant a decrease of 18.9%. In general, we can say that bending properties are susceptible to changes in the composition of cellular changes due to thermal modification [15]. Degradation of hemicelluloses by thermal modification results in a fundamental reduction in mechanical properties. The modulus of elasticity is differently affected than the modulus of rupture. Changes in bending strength are mainly influenced by the degradation of hemicelluloses and highly depend on increasing the temperature and time of thermal modification, while for modulus of elasticity it is possible to assume that the changes in crosslinking of the lignin network affect the rigid structure around the cellulose microfibrils/fibrils and the strength characteristics of the middle lamella. Furthermore, heat-treated wood is less hygroscopic than untreated wood and contains a lower amount of bound water in the cell wall, which affects the modulus of elasticity, making the wood less pliable [2,3,16,17].

In Table 2, it is possible to see selected correlation matrices between individual properties, on the basis of which it is possible to see that the most significant correlation was achieved between the limit of proportionality and modulus of rupture (*r* = 0.89), a lower correlation was then achieved when analyzing the dependence between modulus of elasticity and limit of proportionality (*r* = 0.54). The correlation between the change in density due to thermal modification, the modulus of elasticity (*r* = 0.56), and the modulus of rupture (*r* = 0.51) did not reach a high correlation coefficient representing a linear dependence.

### 2.3. Dynamic Bending

In contrast to static bending properties, a completely different trend can be observed when looking at the decrease in impact bending strength in Figure 2b. The most significant decrease in impact bending strength as a result of thermal modification was recorded at 210 °C (3.9 J∙cm^−2^), when the reference test specimens had an impact bending strength of 10.8 J∙cm^−2^. According to the Duncan test, it can be stated that there were no statistically significant differences (*p* = 0.09) between the *IBS* values at 160 °C (5.1 J∙cm^−2^) and 210 °C thermal modification. There were statistically significant changes (*p* ≤ 0.05) for all remaining degrees of thermal modification. The coefficients of variation for *IBS* for all stages of thermal modification reached relatively high values in the range of 0.41–0.52. The decrease in *IBS* can be explained by the increase in crystalline cellulose in the initial stages of thermal modification, which increases the stiffness of cellulose fibrils (positive effect for modulus of elasticity); however, on the other hand, this effect also contributes to an increase in the brittleness of thermally modified wood [16,17], when this effect can be seen in the change in *IBS*.

Table 3 shows the correlation matrices and their significance levels for *IBS*. A relatively strong dependence of *IBS* was manifested on individual degrees of thermal modification (*r* = 0.72). The correlation coefficient between density and *IBS* did not reach such high values (*r* = 0.53), but even in this case it was a statistically significant correlation.

### 2.4. DSC Thermogram

The differential scanning calorimeter (DSC) analysis was performed due to the change in the correct choice of thermal radiation temperature at which chemical changes occur in untreated black locust wood. Indeed, DSC measurements significantly strengthen our knowledge of the stability of biomacromolecules and macromolecules [18]. The thermogram curve in Figure 3 records the thermophysical properties of the sample. Measurements took place only in the range of thermal modification from 0 to 300 °C. The DSC curve says that first, there is an endothermic reaction where the water evaporates (105–170 °C for black locust). At this stage, the decomposition of hemicelluloses is still negligible. The next stage is the decomposition of hemicelluloses (170–260 °C for black locust). This phase is followed by an exothermic reaction, during which the wood is strongly decomposed and condensable waste products are formed. However, this part is not shown, as a description of higher temperatures was not needed for our research. Because the decomposition of hemicelluloses occurs from 170 to 260 °C, temperatures for thermal modification of 160, 180, and 210 °C were proposed.

### 2.5. Chemical Analysis

The chemical composition of the input raw material has a significant effect on the fiber properties. Table 4 contains the average chemical composition of the investigated black locust wood samples. Chemical representation in woody plants depends on the genotype of the plant, its plant part, place of cultivation, and climate. For this reason, the values in similar studies dealing with the thermal modification of black locust wood may differ.

Table 4 shows that the amount of ash, extractives, cellulose, and holocellulose decreases during thermal modification. At the same time, it is possible to observe a relative increase in lignin content, which is caused by the absence of a correction for the total weight loss during heating. The weight loss increased with increasing thermal modification temperature. The increase in weight loss was 6.8% at 160 °C, 9.4% at 180 °C, and 16.0% at 210 °C thermal modification. A comparison of the measured values for individual wood components with the results of other similar studies is shown in Figure 4 and Figure 5.

Figure 4 and Figure 5 clearly show that black locust (*Robinia pseudoacacia* L.) behaves similarly to *Acacia mangium* [22], but with significantly higher decreases in the case of cellulose and holocellulose. Compared to bamboo [19], another fast-growing plant, our measured values show the most similar trends. Gaff et al. [20] dealt with spruce and European oak. In the case of the amounts of spruce and oak extractives, the values increase with the gradually increasing degree of thermal modification, which is the opposite trend to the investigated black locust. However, they achieved similar values in the case of lignin and for holocellulose above a modification temperature of 200 °C. Differences in results may be due to different climates, plant parts, modification methods, and other factors. In general, except for cellulose, our results of chemical composition after thermal modification are similar to other heat-treated plants.

Table 5 shows the values of hemicelluloses, and specifically pentoses and hexoses. As expected, the content of hemicelluloses decreases with increasing temperature, as they start to decompose around 160 °C. Of course, in the case of pentoses and hexoses, there is also observed a decrease in both fractions. The more significant decrease shows pentoses due to their lower stability compared to hexoses. The proportion of hexoses decreases in the first phase up to 160 °C, and then between 160 and 210 °C is already relatively constant. Another significant decrease can only be expected at higher modification temperatures. The results of the representation of hemicelluloses in the investigated black locust are shown in Figure 6, even in comparison with the results of other similar studies published by other authors.

Even after thermal modification at a temperature of 210 °C, black locust wood contains about 25.3% hemicelluloses, which is more than the 24.7% reported by Wahab et al. [21]. Degradation of hemicelluloses is more significant with increasing temperature of thermal modification. Hemicelluloses of black locust already started to degrade in our case at 160 °C, but Razak et al. [24] reported even higher decomposition degradation up at 180 °C. This degradation is influenced by forming formic and acetic acids, which arise mainly from O-acetyl-galacto-glucomannan under the influence of elevated temperature and thereby cause acid-catalyzed degradation of polysaccharides [25]. Therefore, the degree of degradation of hemicelluloses during thermal modification is proportional to the decrease in acetyl groups [22]. Although analysis of individual hemicellulose monomers was not carried out, a more significant decrease occurs in pentoses (xylose and arabinose), which is also stated in the study by Candalier et al. [26]. Similar results were published achieved by Čabalová et al. [27]. At 210 °C, they observed a decrease in the case of pentoses of 51.3%, and decrease in hexoses of 7.0%. In our case, at the same temperature of thermal modification, we measured decrease in pentoses of 52.5%, and in the case of hexoses, the decrease was 19.2%. This higher decrease could be caused precisely by the use of black locust wood with a different hemicellulose structure than that of Norway spruce [27]. By comparing the previous findings with the experimental results in this study, it can be stated that the stability of cellulose was greater than that of hemicelluloses, which results from many published works of [2,23,25,26,28,29,30,31].

### 2.6. Cellulose Degree of Polymerization

The degree of polymerization was determined viscometrically using the solvent FeTNa, sodium tartrate with iron complexes. The results of the degree of polymerization determined for alfa-cellulose isolated from each thermally modified and unmodified series of investigated samples are shown in Table 6.

However, these values may differ, because most authors, e.g., Kubovský et al. [32], analyzed the degree of cellulose polymerization from isolated cellulose, while we analyzed it from alpha-cellulose. Alpha-cellulose was used because it has longer chains than Seifert cellulose. It is the insoluble part of cellulose in alkali.

At a temperature of 210 °C, the degree of polymerization decreases, which is due to the cleavage of the low-molecular-weight chain when exposed to temperatures above 200 °C. Thermooxidative homolytic radical depolymerization reactions take place more significantly at temperatures above 150 °C. In the first stage of these reactions, free radicals are formed, which in the second stage react with oxygen to form peroxide radicals (•OH). After reorganization of bonds, lactones are formed. Lactones of other oxidized forms of cellulose (e.g., –CH=O, >C=O, –COOH) are decomposed by decarbonylation and decarboxylation with the formation of CO and CO_2_, but also by dehydration with the formation of H_2_O. Thermal oxidation and dehydration are controlled by diffusion processes and take place mainly in the amorphous portions of polysaccharides. Statistical degradation of the cellulose chain probably occurs by the action of water under the catalytic effect of a proton.

### 2.7. Cellulose Crystallinity

The crystallinity index is a commonly used parameter for quantifying the crystalline part in cellulosic materials. In our case of analysis, infrared spectroscopy with Fournier transformation (FT-IR) was used, where the crystallinity index was determined from the relative heights of the peaks, similarly to the authors of Park et al. [33] and Tribulová et al. [34]. The specific absorption bands, or characteristic bands of this bonds, are shown in Figure 7 for black locust wood.

Figure 7 shows individual FT-IR spectra of thermally modified black locust cellulose samples. A broad absorption band around 3400 cm^−1^ appears during polysaccharides O–H valence vibrations [32,33,35,36]. In our case, when analyzing a thermally modified sample, there is a slight decrease in absorbance at this band due to reduction in free hydroxyl groups compared to the standard untreated wood sample. The bands of the measured spectra with a wavenumber of 2895 cm^−1^, the region of C–H valence vibrations in methylene groups, have absorbance values almost unchanged depending on the thermal modification at samples modified at lower temperature, and at temperature of 210 °C there is observed a slight decrease [36]. This region (1500–1200 cm^−1^) is considered to be the “local symmetry” region, which mainly includes deformation vibrations of groups with local symmetry, such as is CH_2_, and the numerous C-OH deformations found in carbohydrates [37]. However, due to the overlapping of different bands of different vibrations, this region is quite crowded and assigning the observed bands by classical group–frequency correlations is challenging [36]. Nevertheless, it is possible to say that these changes will be related to the increasing proportion of the crystalline region. The band around 1429 cm^−1^ shows an increase in absorbance due to increasing temperature. Kubovsky et al. [32] state that this may be evidence of an increase in the amount of crystalline cellulose. The absorption band at 1368 cm^−1^, CH_2_ deformation vibration in cellulose and hemicelluloses, shows a slight increase due to the thermal modification temperature. Hemicelluloses degrade at the temperature range of 180–350 °C, so there are still some present. The band around 1315 cm^−1^ refers to the crystallinity of cellulose, where absorbance also increases due to thermal modification. The intensity at 1102 cm^−1^ decreased for the sample treated at the maximum temperature. The band around 897 cm^−1^ gradually increased with increasing modification temperature.

Bands at 1160 cm^−1^ to 1030 cm^−1^ are characteristic for hemicelluloses and celluloses, and are assigned to C-O-C deformation or valence vibrations [35]. Different glycosidic bond configurations also lead to differences in the 1000–920 cm^−1^ region. A slight decrease in its intensity for the sample modified at a temperature of 210 °C may be due to thermal degradation of β-(1,4) glycosidic linkages. A decrease in absorption in this band indicates a decrease in the amorphous form of cellulose. The band at 897 cm^−1^ is specific to the valence vibration of the glucose ring [38].

Total crystallinity index (*TCI*) and lateral order index (*LOI*) were determined from FT-IR spectra of cellulose. These parameters were used to determine changes in cellulose crystallinity in thermally modified wood compared to the reference sample that was not thermally treated. *TCI* is proportional to the overall degree of cellulose crystallinity in the wood, and *LOI* is correlated with the overall degree of ordering in the cellulose. Results for the black locust wood samples are shown in Table 7.

The *TCI* values, when comparing the results, increase with the gradual increase in temperature. The increase in cellulose crystallinity is influenced by the degradation of hemicelluloses and less-ordered amorphous cellulose. That is why we usually talk about the so-called “relative” increase.

During the initial stages of thermal modification, the degree of crystallinity increases, but as the heating time increases, thermal decomposition of cellulose occurs, which, on the contrary, is accompanied by its decrease. Nakao et al. [39] found an increase in crystallinity during short heating times. Kubojima et al. [40] heated spruce wood at temperatures from 120 °C to 200 °C in air and nitrogen. Cellulose crystallite width and crystallinity increased during the initial stage when heated in the range of 120–160 °C, then settled down. At higher temperatures, this initial increase was followed by a gradual decrease with increasing heating time. A higher increase in crystallinity was discovered with wood samples heated in a moist environment, but this was not found to be the case in native cellulose [3]. Other components of the cell wall, such as xylose and mannose, which are not degraded during thermal modification, contribute to the increase in crystallinity. Li et al. [41] studied steam-heat-treated teak wood using FT-IR and observed an increase in the vibrational motion of the glucose ring, probably due to cleavage and dehydration of amorphous carbohydrates and/or crystallization of the paracrystalline region of cellulose, which may cause an increase in the proportion of crystalline cellulose [42]. Kačíková et al. [43] reported values for *TCI* for thermally modified spruce as 2.38 (reference sample), 2.50 (158 °C), and 2.56 (221 °C), while at 251 °C, the resulting value was 5.00. According to the values, we can observe that the total crystallinity index in this case increases only slightly up to a temperature of 220 °C, and a significant increase occurs only after exceeding this limit. Lopes et al. [42] modified teak wood at 180 and 200 °C. The resulting *TCI* value of the reference sample was 2.78; at 180 °C, 2.79; and at 200 °C, the resulting value was 3.32. The resulting *TCI* values in this work correlate with the values of the mentioned measurements and do not show significant fluctuations.

Higher *LOI* values were observed in samples subjected to higher thermal modification temperatures. The increase in *LOI* at 160 °C and 180 °C is negligible, but at 210 °C, it becomes more significant and confirms the increasing trend. The increase in the index indicated that at 210 °C, the cellulose underwent crystallization. The recorded increase in the values of this index results from a greater susceptibility to the degradation of the amorphous regions of cellulose. When comparing the resulting values with other authors, it was confirmed that the *LOI* increases with increasing temperature. For example, at thermally modified spruce: 1.15 (reference sample), 2.50 (158 °C), 2.56 (221 °C) [36]; at heat-treated teak wood: 0.13 (ref. sample), 0.15 (180 °C), and 0.17 (200 °C) [23]. The results show that all *TCI* and *LOI* values increased slightly in cases of modification temperatures 160 °C and 180 °C and more significantly at 210 °C compared to the reference sample. This increase in crystallinity could also be due to the recrystallization of amorphous regions due to rearrangement or reorientation of cellulose within these regions. The higher crystallinity of cellulose can be caused by the already-mentioned thermal modification, which leads to a reduction in the amorphous fraction [1].

The ratio between the absorbance of the bands at 3332 and 1320 cm^−1^ was used to determine the *HBI* values. The ratio between these bands represents the amount of bound water in the fiber structure [28]. The results of our measurement show a presumed decrease in these values with increasing modification temperature. The ratio between these bands at 3332 and 1320 cm^−1^ represents the amount of bound water in the fiber structure. When wood is subjected to thermal modification, molecular water evaporates, and subsequently, due to the degradation of hemicelluloses and the recrystallization of amorphous portions of cellulose, the proportion of free hydroxyl groups also decreases. Yuan et al. [35] reported that this decrease in values could be related to the fact that with thermal oxidation, more oxygen-containing groups were formed, and the decomposition of water promoted the modification of the molecular structure to form new hydrogen bonds in the crystalline cellulose structure.

## 3. Material and Methods

### 3.1. Materials

The input material for test specimens was black locust wood (*Robinia pseudoacacia* L.); the test specimens for testing both static and dynamic bending had dimensions of 300 × 20 × 20 mm (axial × radial × tangential). WoodEye and X-ray multispectral scanner was used for sample selection Microtec (Brixen, Italy). The scanning device was used to separate wood with similar structural features, specifically to reveal internal cracks, knots, and so on. Before the actual testing, the test samples were conditioned to unify the moisture content to 12% (±1%) and then heat-treated with steam at temperatures of 160, 180, and 210 °C; more detailed information is given in chapter 2.2. The air conditioning was carried out at a relative humidity of 65% and a temperature of 20 °C to constant weight before each measurement of mechanical properties. The moisture content and density of all test specimens before and after thermal modification were measured according to ISO 13061-1 and ISO 13061-2 standards.

The samples before thermal modification were subjected to temperature spectrum measurements using differential scanning calorimetry (DSC) on a DSC 3+ device Mettler Toledo (Greifensee, Switzerland). This measurement was made due to the selection of the thermal modification temperature at which chemical changes occur.

### 3.2. Thermal Modification

The standard thermal modification method, known as the ThermoWood^®^ (Helsinki, Finland) process, was used for the research. The experimental thermal chamber from the company Katres using an atmosphere as a protection medium was used for the modification. The process of using thermal modification can be divided into 3 phases. First, there was a phase of drying and wood heating to the set temperature (160, 180, and 210 °C) with a heating increase of 8 °C per hour to ensure heating of the wood in the entire cross section; in this stage, heated steam was also applied. In the second phase, the thermal modification took place for 3 h (while maintaining steam level). The last step was cooling the test samples, gradually reducing the temperature in the chamber, and moistening the wood to achieve a wood moisture content in the range of 5–7% [44].

### 3.3. Experimental Methods

#### 3.3.1. Mechanical Analysis

According to the EN 310 standard, the three-point bending method was used for static bending. A universal testing device FPZ 100-Tira (Schalkau, Germany) was used for the test. The loading force was applied in the middle of the tested specimen in the tangential direction with a loading speed of 3 mm∙min^−1^, with a set distance of the lower supports of 240 mm. The observed bending characteristics were calculated from the resulting force-deflection diagrams (Equations (1)–(3)):(1)$$MOE=\frac{\left(F_{2}-F_{1}\right)l_{0}{}^{3}}{4bh^{3}\left(y_{2}-y_{1}\right)}$$
(2)$$LOP=\frac{3F_{E}l_{0}}{2bh^{2}}$$
(3)$$MOR=\frac{3F_{max}l_{0}}{2bh^{2}}$$
where *MOE* represents modulus of elasticity (MPa), (*F*_2_ − *F*_1_) is the difference between the force at 40% and 10% of loading (N), (*y*_2_ − *y*_1_) is the difference between the deflection at 40% and 10% of loading (mm), b and h represent dimensions of the cross section of test samples (mm), *LOP* represents the limit of proportionality (MPa), *F*_E_ is the force at the limit of proportionality (N), *MOR* represents the modulus of rupture (MPa), Fmax is the force at the maximum breaking point of the test sample (N), *l*_0_ is the distance between the supports (mm).

To determine the impact bending strength (dynamic bending), a test was performed using a Charpy hammer with a hammer weight of 20 kg. The direction of loading of the test specimens was in the tangential direction. Similar to static bending, the spans for the test specimens were set to 240 mm. The impact bending strength was calculated using Equation (4) according to ISO 3348 standard.
(4)$$A=\frac{Q}{bh}$$
where *A* represents the impact bending strength of wood (J∙cm^−2^), *Q* is the work needed for breaking test samples by Charpy hammer (J), *b* and *h* represent the dimensions of the cross section of test samples (cm).

#### 3.3.2. Chemical Analysis

A chemical analysis of the essential components was carried out for black locust (*Robinia pseudoacacia* L.) samples. The samples were always for an individual batch—random for reference and thermal modification at 160 °C, 180 °C, and 210 °C. Random samples were first ground using a knife mill IKA MF 10 BASIC (Staufen, Germany) and passed through sieves. For chemical analyses, the fraction that fell through the 3.15 mm mesh and was retained on the 0.25 mm mesh was used. The samples were first analyzed for the inorganic content in the form of ash according to TAPPI T 211 om-02. The extractive organic part was determined according to TAPPI T 204 cm-97 by extraction using a Soxhlet apparatus when a mixture of ethanol and toluene in a ratio of 7:3 was used as the extraction solvent. Seifert cellulose [45], Klason lignin using sulfuric acid according to TAPPI T 222 om-01, and holocellulose according to Wise [46] were determined from the extracted remaining part. Using the amount of holocellulose and cellulose, the content of hemicelluloses was approximately calculated. The last determined component was the content of pentoses according to TAPPI T 19 wd-71, from which the content of hexoses was also derived.

In addition to the chemical analysis of the main wood components, the degree of polymerization of alpha-cellulose according to TAPPI T230 om-08 was determined viscometrically using the limiting viscosity number. The degree of polymerization was determined using FeTNa solvent—sodium tartrate with iron complexes [47].

As the last part of the chemical analysis, the FTIR spectra of Seifert cellulose were analyzed using the Nicolet iS20 spectrophotometer (Waltham, MA, USA). The increase in the crystallinity of wood cellulose by heat treatment has been reported by many studies [48,49,50]. The crystalline state significantly influences other properties such as the elasticity, absorptive capacity, and other industrially valuable physical properties of the fiber. Therefore, the objective of this study was to determine the crystalline behavior such as the degree of crystallinity. Cellulose pellets of 1.2 cm diameter and about 1 mm thickness were prepared by pressing (50 MPa) of ca. 50 mg of sample using by CrushIR digital hydraulic press PIKE Technologies (Fitchburg, MA, USA). Spectra were obtained by accumulating 64 interferograms with a resolution of 4 cm^−1^ in absorbance mode at wavenumbers in the 400–4000 cm^−1^. All spectra were baseline-corrected, assigned to the C–O stretching vibration in cellulose, and thus normalized to the 1032 cm^−1^ band, and the average of multiple spectra was used as the representative plot for each sample. After deconvolution, several parameters, total crystallinity index (*TCI*), lateral order index (*LOI*), and hydrogen bond intensity (*HBI*), could be calculated and compared [34,51]. The above-mentioned parameters were calculated as the proportion of absorbance peaks at the absorbances given in Equations (5)–(7).
(5)$$TCI=\frac{A_{1368}}{A_{2894}}$$
(6)$$LOI=\frac{A_{1429}}{A_{897}}$$
(7)$$HBI=\frac{A_{3332}}{A_{1320}}$$

## 4. Conclusions

The process of thermal modification of black locust wood caused a decrease in the strength static bending properties (*MOR*, *LOP*, *MOE*), which decreased with increasing temperature of thermal modification. This is due to the degradation of cellulose, which is responsible for the strength of the wood. However, during thermal modification, the degradation of the cell walls of the polymers is an advantage, as there is a reduction in the swelling of the thermally treated wood. By reducing the amorphous part of the polysaccharides present in wood, the rot resistance and dimensional stability of thermally treated wood are improved.

The dynamic bending properties of *IBS* are significantly reduced already at lower temperatures of thermal modification due to an increase in crystalline cellulose in the initial stages of thermal modification, which causes an increase in wood brittleness, whereas the static bending properties (*MOE*, *LOP,* and *MOR*) reach significant decreases at higher degrees of thermal modification. At the same time, the elastic modulus does not have such a significant reduction at lower temperatures, but it is already very strongly affected at a temperature of 210 °C. The reduction in bending strength strongly depends on the degradation processes of hemicelluloses, which degrade to a greater extent with increasing temperature. These changes of static bending properties are also influenced by the decrease in the *DP* of cellulose at temperatures above 200 °C.

Although the heat treatment caused the depolymerization of cellulose, degradation of hemicelluloses, and reduction of mechanical strength of wood depending on the thermal degradation level, the modification has its advantages. The presented results of our work show similar values that were measured at thermally modified spruce and oak wood. Therefore, thermal modification is a possible solution for the industrial processing and wider application of black locust wood.

## Figures and Tables

**Figure 1 ijms-23-15652-f001:**
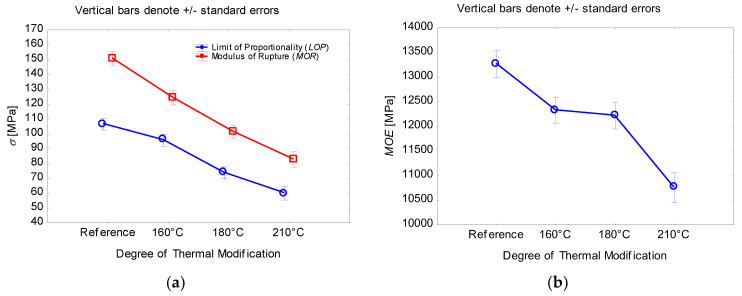
(**a**) The effect of thermal modification on the *LOP* and *MOR* values; (**b**) the effect of thermal modification on the *MOE* values.

**Figure 2 ijms-23-15652-f002:**
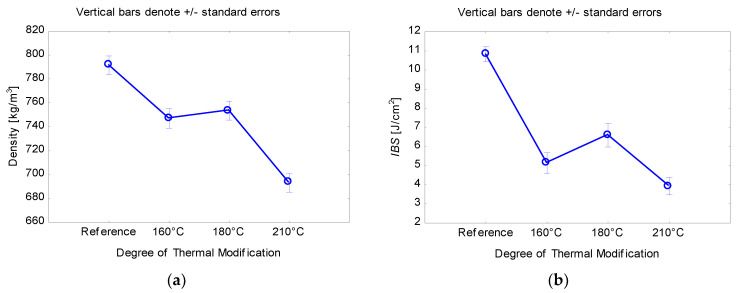
(**a**) Influence of thermal modification on density; (**b**) influence of thermal modification on impact bending strength.

**Figure 3 ijms-23-15652-f003:**
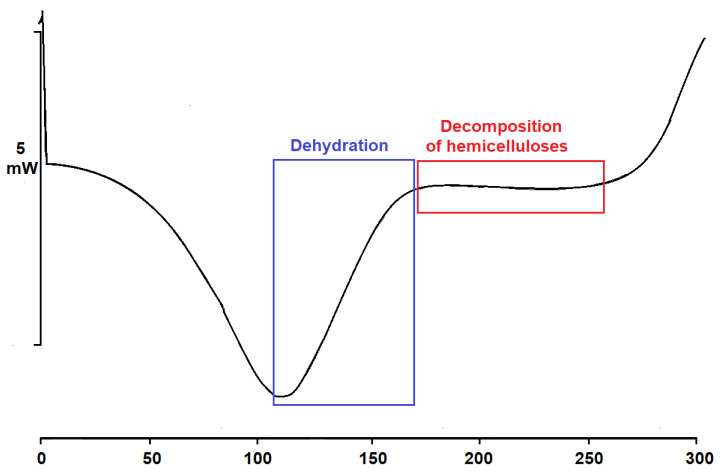
Effect of temperature on chemical reactions of black locust wood.

**Figure 4 ijms-23-15652-f004:**
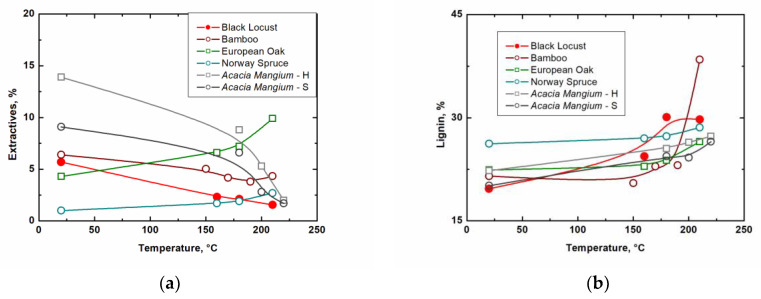
(**a**) Influence of thermal modification on extractives; (**b**) influence of thermal modification on lignin (bamboo [19], European oak, Norway spruce [20], *Acacia mangium*—heartwood, sapwood [21]).

**Figure 5 ijms-23-15652-f005:**
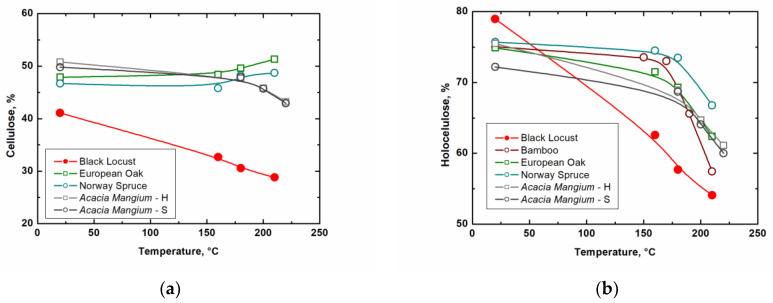
(**a**) Influence of thermal modification on cellulose; (**b**) influence of thermal modification on holocellulose (bamboo [19], European oak, Norway spruce [20], *Acacia mangium*—heartwood, sapwood [21]).

**Figure 6 ijms-23-15652-f006:**
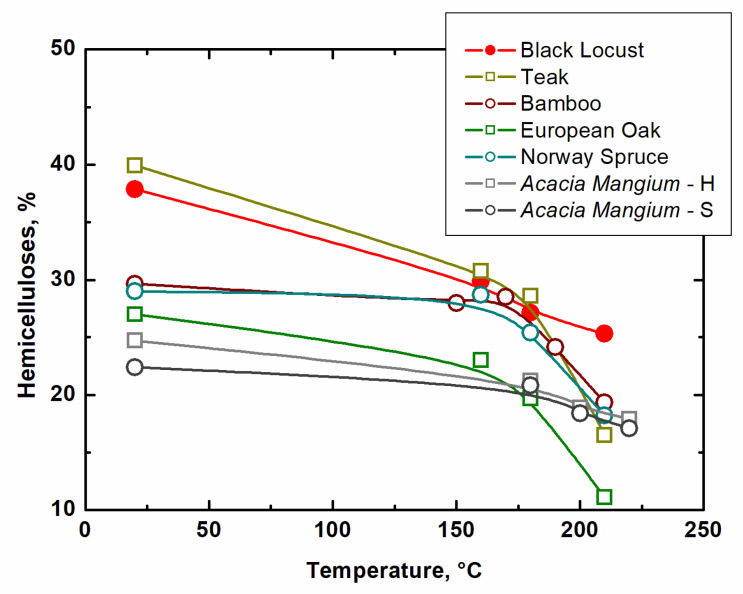
Influence of thermal modification on hemicelluloses (teak [23], bamboo [19], European oak, Norway spruce [20], *Acacia mangium*—heartwood, sapwood [21]).

**Figure 7 ijms-23-15652-f007:**
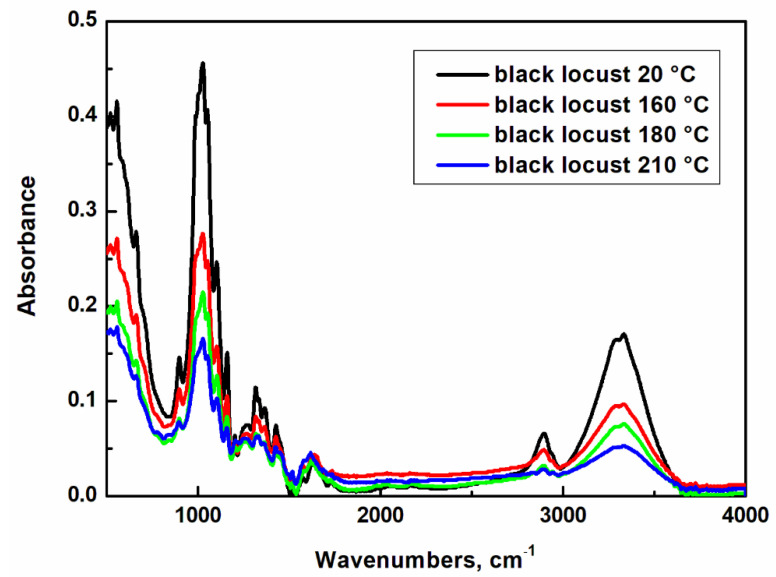
The region of FT-IR spectra of cellulose with the crystallinity-related bands corresponding to reference and thermally modified samples after being isolated from black locust wood.

**Table 1 ijms-23-15652-t001:** Changes of selected static and dynamic properties.

Temperature, °C	Density, kg·m^−3^	Static Bending			Dynamic Bending
		Modulus of Elasticity, MPa	Limit of Proportionality, MPa	Modulus of Rupture, MPa	Impact Bending Strength, J·cm^−2^
Unmodified (reference)	790.2 (4.5)	13,269.4 (8.8)	106.6 (20.0)	150.8 (15.4)	10.8 (16.2)
160	751.6 (4.9)	12,324.6 (11.1)	95.7 (19.5)	124.6 (21.2)	5.1 (45.1)
180	755.4 (4.5)	12,216.2 (7.7)	73.7 (27.3)	101.4 (18.6)	6.6 (41.3)
210	686.8 (5.1)	10,755.3 (16.4)	60.0 (28.1)	82.8 (22.8)	3.9 (52.5)

Values in parenthesis are coefficients of variability.

**Table 2 ijms-23-15652-t002:** Absolute correlations and their *p*-values for static bending.

Variable	Degree of Thermal Modification	Density	*MOE*	*LOP*	*MOR*
Degree of thermal modification	–	0.565 (0.000)	0.469 (0.000)	0.573 (0.000)	0.692 (0.000)
Density	0.565 (0.000)	–	0.563 (0.000)	0.440 (0.000)	0.513 (0.000)
*MOE*	0.469 (0.000)	0.563 (0.000)	–	0.547 (0.000)	0.654 (0.000)
*LOP*	0.573 (0.000)	0.440 (0.000)	0.547 (0.000)	–	0.890 (0.000)
*MOR*	0.692 (0.000)	0.513 (0.000)	0.654 (0.000)	0.890 (0.000)	–

Values in parenthesis are coefficients of variability.

**Table 3 ijms-23-15652-t003:** Absolute correlations and their *p*-values for dynamic bending.

Variable	Degree of Thermal Modification	Density	Energy for *IBS*	*IBS*
Degree of thermal modification	–	0.623 (0.000)	0.732 (0.000)	0.728 (0.000)
Density	0.623 (0.000)	–	0.522 (0.000)	0.534 (0.000)
Energy for *IBS*	0.732 (0.000)	0.522 (0.000)	–	0.999 (0.000)
*IBS*	0.728 (0.000)	0.534 (0.000)	0.999 (0.000)	–

Values in parenthesis are coefficients of variability.

**Table 4 ijms-23-15652-t004:** Chemical composition (in mass % of oven dried samples).

Sample/Wood Component	Ash	Extractives Ethanol-Toluene	Lignin	Cellulose	Holocellulose
Reference	1.87 (0.000)	5.69 (0.757)	19.66 (1.524)	41.10 (0.191)	78.98 (0.249)
160 °C	1.73 (0.000)	2.31 (0.074)	24.38(5.089)	32.69 (1.467)	62.58 (1.609)
180 °C	1.59 (0.000)	2.11 (0.046)	30.10 (3.408)	30.57 (1.566)	57.70 (0.051)
210 °C	1.33 (0.000)	1.55 (0.135)	29.75 (1.825)	28.80 (0.712)	54.10 (0.133)

Standard deviation values are in parenthesis.

**Table 5 ijms-23-15652-t005:** Hemicelluloses (in mass % of oven dried samples).

Sample/ Components	Hemicellulose	Pentoses	Hexoses
Reference	37.88 (0.185)	15.99 (0.054)	21.89 (0.155)
160 °C	29.88 (1.427)	12.41 (0.345)	17.48 (1.274)
180 °C	27.13 (1.566)	9.71 (0.009)	17.43 (1.566)
210 °C	25.30 (0.711)	7.60 (0.021)	17.70 (0.709)

Standard deviation values are in parentheses.

**Table 6 ijms-23-15652-t006:** Cellulose degree of polymerization.

Sample/Modification Temperature	Reference	160 °C	180 °C	210 °C
black locust	280 (2)	311 (2)	347 (2)	302 (3)

Standard deviation values are in parenthesis.

**Table 7 ijms-23-15652-t007:** Crystallinity characteristics of isolated cellulose.

Sample/Parameter for Crystallinity Characterization	*TCI*	*LOI*	*HBI*
Reference	1.406	0.507	1.528
160 °C	1.494	0.547	1.167
180 °C	1.742	0.588	1.156
210 °C	1.911	0.649	0.845

*TCI*—total crystallinity index, *LCI*—lateral order index, *HBI*—hydrogen bond intensity

## Data Availability

Data are available on request due to ethical restrictions. The data presented in this study are available on request from the corresponding author. The data are not publicly available due to (unfinished research).

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
