# Peer review of "Degradation of Chemical Components of Thermally Modified Robinia pseudoacacia L. Wood and Its Effect on the Change in Mechanical Properties"

_ijms, 2022, doi:10.3390/ijms232415652_

Round 1

Reviewer 1 Report

It should be emphasized that proper citation is of great importance in academic writing for it enables knowledge accumulation and maintains academic integrity. Some important references are missing.

Scientific names of species should be italicized.

Figures are of too low resolution

It is not indicated the importance of study and application to practice

Author Response

Reviewer 1: In the attachment.

Dear Reviewer no. 1,

Thank you very much for reviewing our post and for providing additional recommendations. We have tried to follow them all to improve the quality of our posts.

The manuscript is focused on an important research topic: The degradation of Chemical Components of Thermally Modified Robinia Pseudoacacia Wood and Its Effect on the Change of Mechanical Properties.

Based on your recommendations, you will see that we have adjusted the citations and scientific names and modified the graphs to make them more readable. Overall, we have tried to convert the entire manuscript into a more acceptable form.

Regards,

The Authors

Reviewer 2 Report

The manuscript „Degradation of Chemical Components of Thermally Modified Robinia Pseudoacacia Wood and Its Effect on the Change of Mechanical Properties” describes the effect of modification at three different temperatures on the chemical composition and mechanical parameters of black locust wood. The research is interesting and important, and several experiments have been performed. However, it lacks a scientific discussion of the obtained results, there are no conclusions from the study, and technical or experimental details are missing in several places. Also, the article includes many generic or vague statements and information, while in a scientific paper, scientific facts are required. Therefore, I cannot recommend the manuscript for publication in this form and suggest its thorough correction and supplementation with relevant information. More detailed comments, suggestions, and questions can be found in the attached pdf file.

Author Response

Reviewer 2: In the attachment.

Dear Reviewer no. 2,

Thank you very much for reviewing our post and for providing additional recommendations. We have tried to follow them all to improve the quality of our posts.

The manuscript is focused on an important research topic: The degradation of Chemical Components of Thermally Modified Robinia Pseudoacacia Wood and Its Effect on the Change of Mechanical Properties.

A scientific discussion of the obtained results was added to the manuscript. The conclusion has also been redone. Overall, we have tried to convert the entire manuscript into a more acceptable form.

Regards,

The Authors

Round 2

Reviewer 2 Report

The manuscript has been partially corrected and supplemented, but still needs some clarifications and supplementations. It lacks conclusions and this part is mandatory for a scientific paper. Detailed comments can be found in the attached pdf file.

Author Response

Reviewer 2: In the attachment.

Dear Reviewer no. 2,

Thank you very much for reviewing our post and for providing additional recommendations. We have tried to follow them all to improve the quality of our posts.

You will see that we have better described the purpose of the research based on your recommendations.

The abstract and conclusion have also been corrected. The conclusion should now be more concise.

Regards,

The Authors
